# Determining how biotic and abiotic variables affect the shell condition and parameters of *Heliconoides inflatus* pteropods in the Cariaco Basin

Rosie L. Oakes[1] and Jocelyn A. Sessa[1]

[1]Academy of Natural Sciences of Drexel University, Philadelphia, PA 19103, USA

*Correspondence to*: Rosie L. Oakes (roakes@drexel.edu)

**Abstract**

Pteropods have been nicknamed the 'canary in the coal mine' for ocean acidification because they are predicted to be among the first organisms to be affected by changing ocean chemistry. This is due to their fragile, aragonitic shells and high abundances in polar and sub-polar regions where the impacts of ocean acidification are most pronounced. For pteropods to be used most effectively as indicators of ocean acidification, the biotic and abiotic factors influencing their shell formation and dissolution in the modern ocean need to be quantified and understood. Here, we measured the shell condition (i.e., the degree to which a shell has dissolved) and shell characteristics, including size, number of whorls, shell thickness, and shell volume (i.e., amount of shell material) of nearly fifty specimens of the pteropod species *Heliconoides inflatus* sampled from a sediment trap in the Cariaco Basin, Venezuela over an 11-month period. The shell condition of pteropods from sediment traps have the potential to be altered at three stages: 1) when the organisms are live in the water column associated with ocean acidification, 2) when organisms are dead in the water column associated with biotic decay of organic matter and/ or abiotic dissolution associated with ocean acidification, and 3) when organisms are in the closed sediment trap cup associated with the abiotic alteration by the preservation solution. Shell condition was assessed using two methods: the *Limacina* Dissolution Index (LDX) and the opacity method. The opacity method was found to capture changes in shell condition only in the early stages of dissolution, whereas the LDX recorded dissolution changes over a much larger range. Because the water in the Cariaco Basin is supersaturated with respect to aragonite year-round, we assume no dissolution occurred during life, and there is no evidence that shell condition deteriorated with the length of time in the sediment trap. Light microscope and SEM images show the majority of alteration happened to dead pteropods while in the water column associated with the decay of organic matter. The most altered shells occurred in samples collected in September and October when water temperatures were warmest, and the amount of organic matter degradation, both within the shells of dead specimens and in the water column, was likely to have been the greatest.

Changes in the hydrography and chemical properties in the Cariaco Basin vary seasonally due to the movement of the Inter Tropical Convergence Zone (ITCZ). Shells of *H. inflatus* varied in size, number of whorls, and thickness throughout the year. There was not a strong correlation between the number of whorls and the shell diameter, suggesting that shell growth is plastic. *H. inflatus* formed shells that were 40% thicker and 20% larger in diameter during nutrient rich, upwelling times when food supply was abundant, indicating that shell growth in this aragonite-supersaturated basin is controlled by food availability. This study produces a baseline dataset of the variability in shell characteristics of *H. inflatus* pteropods in the Cariaco Basin and documents the controls on alteration of specimens captured via sediment traps. The methodology outlined for assessing shell parameters establishes a protocol for generating similar baseline records for pteropod populations globally.

# 1 Introduction

The global ocean has absorbed over a third of anthropogenic carbon dioxide emissions since the industrial revolution (Gruber et al., 2009; Sabine et al., 2004). This has caused the chemistry of the oceans to change, decreasing both the pH and the concentration of carbonate ions in seawater. The impact of this decrease in carbonate ion concentration on mineral formation can be expressed using the saturation state equation of Broecker and Peng (1982):

$$\Omega = \frac{[Ca^{2+}]_{SW} \times [CO_3^{2-}]_{SW}}{[Ca^{2+}]_{saturation} \times [CO_3^{2-}]_{saturation}}$$

where $\Omega$ is the calculated saturation state, $[Ca^{2+}]$ is the concentration of calcium ions, $[CO_3^{2-}]$ is the concentration of carbonate ions, and $SW$ is seawater. At $\Omega$ values greater than one, the seawater is supersaturated with respect to the mineral, and at values less than one, seawater is undersaturated with respect to the mineral, causing it to be chemically unstable.

Recent studies have proposed that biological indicators of carbonate undersaturated waters can be used to monitor future changes in ocean chemistry (Bednaršek et al., 2017, 2019; Gaylord et al., 2018; Marshall et al., 2019). Establishing biological indicators is complicated because organisms are exposed to a multitude of variability in oceanic conditions, from temperature and salinity to carbonate saturation levels and nutrient concentrations, on diurnal, seasonal, and annual timescales. All of these variables have been shown to impact shell growth in calcareous organisms (e.g. Comeau et al., 2009, 2010; Hettinger et al., 2013; Hiebenthal et al., 2011; Joubert et al., 2014; Meinecke and Wefer, 1990; Melzner et al., 2011) and it is therefore crucial that the natural variability of organisms' shell parameters in response to environmental fluctuations is understood prior to their use as indicators of changes in ocean chemistry.

## 1.1 Understanding natural pteropod variability

Pteropods are a group of pelagic molluscs that have been proposed as biological indicators of ocean acidification (Bednaršek et al. 2014a, 2017, 2019). They form their thin (10–15 µm) shells from the mineral aragonite, a more soluble form of calcium carbonate (Mucci, 1983), and therefore are at a greater risk from ocean acidification than organisms with calcitic shells (Fabry, 2008; Orr et al., 2005). Pteropods are protandric hermaphrodites, meaning they transition from juveniles, to mature males, to females during ontogeny (Lalli and Wells, 1978). Their lifespans are thought to be between 0.5 and 2 years (Gannefors et al., 2005; Hunt et al., 2008; Kobayashi, 1974; Wang et al., 2017; Wells, 1976a). Isotopic studies have found that pteropods calcify between 50 and 650 m depth (Fabry and Deuser, 1992; Juranek et al., 2003; Keul et al., 2017) suggesting they are exposed to a wide range of water chemistries during their diurnal migration. Pteropods are also key components of the marine food web, feeding on phytoplankton and small zooplankton, such as diatoms, dinoflagellates, and tintinnids (Gilmer and Harbison, 1986, 1991; Lalli and Gilmer, 1989), and being consumed by zooplankton, krill, fish, and

seabirds (Doubleday and Hopcroft, 2014; Foster and Montgomery, 1993; Hunt et al., 2008; Karnovsky et al., 2008; Pakhomov et al., 1996; Willette et al., 2001).

Because of their sensitivity to ocean acidification, there has been a significant increase in research on this group over the past decades, including incubation experiments, studies on natural $CO_2$ gradients, and descriptions of the genetic variability within natural populations (c.f. Manno et al., 2017). The impact of predicted future conditions on live specimens has been assessed using wide variety of parameters, including calcification (Comeau et al., 2009, 2010; Maas et al., 2018; Moya et al., 2016), shell degradation (Bednaršek et al., 2012b; Bergan et al., 2017; Lischka and Riebesell, 2012), metabolic rates (Lischka and Riebesell, 2017; Maas et al., 2011; Seibel et al., 2012), respiration (Comeau et al., 2010; Maas et al., 2018; Moya et al., 2016), and gene expression patterns (Koh et al., 2015; Maas et al., 2015, 2018; Moya et al., 2016; Thabet et al., 2017). Generally, previous studies have found that as the aragonite saturation state decreases, pteropod calcification rates decrease (Comeau et al., 2010, 2009; Lischka and Riebesell, 2012). This decreased calcification may be manifested in the formation of smaller, thinner, or more porous shells (Bednaršek et al., 2017, 2019; Roger et al., 2012).

Although much has been learned about the response of pteropods to acidification, there are still fundamental processes that remain incompletely understood, including how shell characteristics, such as shell thickness or shell diameter, change through ontogeny, and whether these parameters are affected by ocean chemistry. This work is hampered because pteropods are difficult to culture (Howes et al., 2014), with only one study reporting successfully rearing a captive generation (Thabet et al., 2015). Understanding how shell shape and size change through ontogeny is instead based on measurements from repetitive sampling of natural populations (Hsiao, 1939; Redfield, 1939; Wells, 1976b), and on the diversity of shells in the sedimentary record (Janssen, 1990).

Pteropod samples can be collected live, using plankton nets, or dead, in sediment traps. Although net catches have the advantage of sampling pteropod populations at the time of collection, they only represent a snapshot in time. Pteropods have patchy distributions (Bednaršek et al., 2012a; Thibodeau and Steinberg, 2018; Wang et al., 2017), and therefore pteropod yields in net samples are highly variable. Sediment traps use a large, upward facing cone to collect the flux of organic and inorganic particles that sink through the water column into collection cups containing preservative. These collection cups are automatically closed and switched out on a regular basis (i.e., every two weeks or every month) which enables the flux of particles in the water column, including dead plankton, to be continuously sampled over a longer period than is possible via net catches. Organisms falling through the water column may decay en-route to the sediment trap, which can cause dissolution in calcareous organisms (Lohmann, 1995; Milliman et al., 1999). In pteropod shells specifically, Oakes et al. (2019a) found the majority of post-mortem dissolution was associated with the biotic decay of organic material on the inside of the shell, and therefore specimens from sediment traps do not perfectly capture in-life shell conditions. A further complication of sediment trap data is that interpretation can be skewed by the presence of 'swimmers', i.e., specimens that

were alive when they entered the trap (Harbison and Gilmer, 1986). This is a particular concern with pteropods as they sink to avoid predation (Harbison and Gilmer, 1986) and therefore may enter into the trap while still alive. Additionally, sediment trap samples can be subjected to alteration in the sediment trap cup, due to decay of the organic matter and degradation associated with the preservation solution. For example, a study by Oakes et al. (2019b) found that when left in mercuric chloride or formalin, the most common solutions used in sediment trap studies (e.g., Collier et al., 2000; Manno et al., 2007; Meinecke and Wefer, 1990; Mohan et al., 2006; Singh and Conan, 2008), pteropod shells underwent dissolution over the study period of 15 months. The condition of shells from sediment traps must, therefore, be interpreted in the context of: water column properties when the individuals are alive, post-mortem decay before specimens reach the sediment trap, and potential breakdown during the time they are in the sediment trap cup.

## 1.2 The CARIACO Time Series

The Cariaco Basin is a tectonic depression on the Venezuelan shelf (Fig. 1) separated from the Caribbean Sea by a shallow sill (~140 m) meaning the deep waters of the basin are permanently anoxic (Muller-Karger et al., 2001). The surface water conditions in the Cariaco Basin vary seasonally with the migration of the Inter Tropical Convergence Zone (ITCZ). During the winter and spring (Dec – Apr), the ITCZ moves south, the Easterly trade winds are strong (> 6 m s$^{-1}$), and Ekman transport causes coastal upwelling, bringing cold, high salinity water to the surface (Astor et al., 2003, 2013). During the summer and fall (Aug – Nov), the ITCZ moves north, causing winds to weaken and rainy conditions to become pervasive; there is no upwelling, and surface waters are warm, oligotrophic, and lower salinity relative to the upwelling season (Astor et al., 2013; Muller-Karger et al., 2019). Organic carbon fluxes in the basin vary in response to these hydrographic changes, with one study reporting a tripling of primary productivity in response to upwelling (Thunell et al., 2000). Diatoms, a known food source for pteropods (Lalli and Gilmer, 1989), contribute to over 50% of this organic carbon flux, with their blooms coinciding with hydrographic and nutrient changes during times of upwelling (Romero et al., 2009).

The CARIACO (Carbon Retention In A Colored Ocean) project was a time-series study that ran from 1995–2017 to measure the relationships among physical and biological processes in the Cariaco Basin, Venezuela. The CARIACO time series coupled bi-weekly sediment trap samples with monthly oceanographic cruises to measure hydrography, nutrient concentrations, and biogeochemical parameters (*c.f.* Muller-Karger et al., 2019). There have been numerous studies of planktic foraminifera from the sediment trap samples, investigating their flux, the variability of assemblages both seasonally and interannually, and their ability to record changes in the oxygen isotopic composition and carbonate chemistry of seawater (e.g., Marshall et al., 2013, 2015; McConnell et al., 2009; Tedesco et al., 2007; Tedesco and Thunell, 2003). Despite this focus on calcareous plankton, there have not been any studies on pteropods from the Cariaco sediment trap records.

The wealth of data collected during the CARIACO time series, and the seasonal variability in water column properties, makes the Cariaco Basin an ideal place to study the abiotic and biotic controls on the shell characteristics of *Heliconoides inflatus* pteropods. Temperature, salinity, nutrient concentrations, and carbonate chemistry of the water column were collected as part of the CARIACO time series. To determine how changes in these water column properties affect the shells of pteropods, we assessed 50 specimens from eight sediment trap samples over an 11-month period, using a combination of light microscopy, scanning electron microscopy, and CT scanning.

To compare among and between pteropod shells from different samples, shell diameter was used as a metric for size, and shell thickness and amount of shell material were used as metrics for calcification. Shell thickness has been used as a metric for calcification in previous studies, initially calculated from point measurements on the shell aperture from scanning electron microscope images (Bednaršek et al., 2014b; Roger et al., 2012), and later measured across entire shells from CT reconstructions (Howes et al., 2017; Oakes et al., 2019a; Peck et al., 2018). Here we use modal shell thickness to compare calcification among samples following the methods of Oakes et al. (2019a). Although this method analyses shell thickness across the entire shell, the final, or body, whorl, composed of the most recently calcified material, is the largest portion of the shell in *Heliconoides inflatus* pteropods (Fig. S1) (Fabry and Deuser, 1992; Keul et al., 2017). This final whorl therefore comprises the majority of the shell volume and hence will dominate the modal shell thickness measurement.

## 2 Materials and Methods

### 2.1 Sediment trap collections and water column properties

The samples for this study come from the CARIACO Time Series trap deployed at 150 m water depth (also known as the Z trap) (10° 30.0' N, 64° 38.5' W) (Fig. 1). Sediments were collected continuously for two-week intervals in collection cups that were filled with a borate-buffered formalin solution prior to trap deployment to preserve the sample (Thunell et al., 2000). There were 13 cups in the trap and the trap was retrieved and redeployed every six months (Thunell et al., 2000). On recovery, the contents of the sediment trap cups were washed and split as described in Thunell et al. (2000) and Tedesco and Thunell (2003). A quarter split was washed over a 150-micron sieve with deionized water. Calcareous plankton were wet-picked, and left to dry in a 40°C oven for 24 hours, before being stored for faunal analysis (E.Tappa, pers.comm.).

### 2.2 Specimen selection

We analyzed 50 specimens of *Heliconoides inflatus* (Mollusca, Gastropoda, Euthecosomata, Limacinidae) from eight collection cup samples spanning March 2013 through February 2014 (Table 1). All pteropod specimens were picked from the washed and dried faunal samples by B. Marshall and C. Davis (University of South Carolina). Light microscope images were used to assess shell condition (Fig. S1), scanning electron microscopy was used to assess the extent to which dissolution had occurred on the internal and external areas of the shell, and CT scans were used to determine shell diameter,

number of whorls, shell thickness, total shell volume. Because *H. inflatus* shells are fragile, they often break at the aperture during collection and processing. Although shell diameter and number of whorls were measured on all CT-scanned specimens, a subset of 29 shells with complete apertures was created for further analyses (Table S1). Specimens are deposited in the Malacology collection at the Academy of Natural Sciences of Drexel University, Philadelphia, PA (ANSP).

Catalogue numbers can be found with sample information in Table 1.

## 2.3 Light microscopy

Forty-nine of the 50 *H. inflatus* shells were imaged under the light microscope in order to assess shell condition (i.e., the degree to which shells have undergone dissolution); one specimen broke after CT scanning and therefore was not imaged via light microscopy. Thirty-eight of these 49 shells were imaged on a Zeiss Stemi 2000-C microscope with a Canon G9 camera

in SCN mode, in the Paleoceanography Lab at the Pennsylvania State University; 11 shells were imaged on a Leica S8APO microscope with a Leica DFC HD Camera at the Academy of Natural Sciences of Drexel University. All images are available in the supplemental materials (Fig. S1).

### 2.3.1 Assessment of shell condition

Dissolution visibly affects the shells of pteropods, altering them from glassy and transparent when pristine, to milky-white,

and then white and opaque as they dissolve (Almogi-Labin et al., 1986). The visible changes in pteropod shells have been used as a metric of dissolution. Here we assess the amount of dissolution the pteropod shells have undergone, hereafter referred to as the shell condition, using two methods: the *Limacina* Dissolution Index, and the opacity method. The *Limacina* Dissolution Index (LDX) was designed to assess the extent of dissolution in pteropods from the fossil record using a scale from 0 (pristine shell) to 5 (highly dissolved shell) based on observations made using a light microscope (Gerhardt et al.,

2000; Gerhardt and Henrich, 2001). The opacity method (Bergan et al., 2017) was designed to quantify small changes in shell dissolution by measuring the greyscale values of light microscope images of a shell relative to a black background to determine how much light is able to pass through the shell. A pristine shell will have a low opacity (~0 – 0.25), as the background will be visible through transparent shell, and a highly altered shell will have a high opacity score (~0.5 – 0.7) as the opaque shell will block light from travelling through the shell. The shells in this study were analyzed by Oakes using

both the LDX and opacity methods.

## 2.4 Scanning Electron Microscopy

A subset of seven specimens, spanning pristine to highly altered shell conditions, were imaged using a scanning electron microscope (SEM) to determine the extent of internal and external shell dissolution. Specimens were imaged using an FEI Quanta 600 ESEM at the Nanoscale Characterization Facility at the Singh Center for Nanotechnology at the University of

Pennsylvania, Philadelphia, USA. Samples were mounted on carbon tape and were imaged uncoated. All specimens were

imaged at 200-, 500-, 50,000-, and 100,000-times magnification on the external wall, and 50,000- and 100,000-times magnification on the internal wall where possible (Fig.2. Fig. S2).

## 2.5 CT scanning

### 2.5.1 CT data collection

Forty-four of the 50 *H. inflatus* specimens were CT scanned (Table 1). The remaining six shells fragmented or broke completely prior to CT scanning. CT scanning was conducted using two different CT scanners due to scanner availability (Table S1). Thirty-one specimens were scanned at General Electric Inspection Technology, Lewistown, PA using a GE phoenix v|tome|x m micro-CT system (General Electric, Fairfield, CT, USA). Specimens were scanned at a resolution of 1–2 µm/voxel using the 180kV nanofocus tube with a diamond target and a beam energy of 65 kV and 230 µA. X-ray radiographs were collected with 500 ms exposure times and five radiographs were collected and averaged (average 5, skip 1) at 1000 projections around the specimen, yielding an overall scan time of 50 minutes. Because of the closure of the GE facility, the remaining 13 specimens were scanned at the Microscopy and Imaging Facility and the American Museum of Natural History, New York, NY using a GE phoenix v|tome|x s 240 dual tube 240/180 kV system (General Electric, Fairfield, CT, USA). Specimens were scanned at a resolution of 1–2 µm/voxel using the 180 kV nanofocus tube with a diamond target and a beam energy of 65 kV and 230 µA. X-ray radiographs were collected with 400 ms exposure times and three radiographs were collected and averaged (average 3, skip 1) at 1500 projections around the specimen yielding an overall scan time of 40 minutes.

Ideally, all scans would have been conducted with the same equipment and parameters, but the GE facility closure, and limited scan time availability at the AMNH, resulted in reducing the total scan time from 50 minutes at the GE facility to 40 minutes at the AMNH in order to scan the greatest number of shells possible. To assess the impact of using both different scanners, and different scan parameters on the calculated modal shell thickness, a key measurement used in this study, one specimen was scanned four times: 1) original scan at GE; 2) scan at AMNH; 3) re-scan at AMNH; 4) rescan at AMNH using scan parameters from GE (Table S2; see supplemental materials for further details). Although there were minor variations among scans (Fig. S3), the modal shell thickness calculated for all four scans was 0.008 mm. This demonstrates that modal shell thickness is a robust metric and was not impacted by the different scanners, scan parameters, or scan times used in this study.

### 2.5.2 CT data processing

All CT data were reconstructed using *datos/x* v. 2 (General Electric, Wunstorf, Germany) and analyzed using *VGStudio MAX* v. 3.1 (Volume Graphics, Heidelberg, Germany). Shell material was differentiated from background using the automatic surface determination module. Some shells were filled with other materials, such as foraminifera tests or sediment. To ensure

that only the shell of the pteropod was analyzed, a region of interest (ROI) was created from the surface and non-pteropod shell material was manually removed from the ROI. The resulting surface was exported as a *.DICOM image stack. The volume, or amount of pteropod shell material, was calculated using the properties tool in VG Studio MAX v. 3.1.

### 2.5.3 Quantifying shell parameters

Data were visualized and measured in Avizo v. 9.4.1. The shell diameter was measured at the widest part of the shell following the methods of Lischka et al., (2011) using the caliper tool in Avizo v. 9.4.1 (Fig. 3). The number of whorls were counted to the nearest eighth of a whorl following the method of Janssen (2007) (Fig. 3). Shell thickness was measured using the BoneJ plugin (Doube et al., 2010; Hildebrand and Rüegsegger, 1997) in ImageJ (Schneider et al., 2012) following the methods of Oakes et al. (2019a).

### 2.6 Seawater Chemistry

Water chemistry was analyzed monthly as part of the Cariaco Basin ocean time series program. These data are publicly available at *http://imars.marine.usf.edu/WebPageData_CARIACO/Master_Hydrography/*. Water samples were collected at discrete depth intervals to measure nutrient concentrations and carbonate chemistry parameters, the details of which can be found in Astor et al. (2011). There are 12 water sampling datasets that span the duration of this study (March 2013 – February 2014). Aragonite saturation ($\Omega_{arag}$) was calculated indirectly from the pH and total alkalinity (TA) data from the timeseries using CO2SYS (Pierrot et al., 2006). Carbonate dissociation constants were used from Mehrbach et al. (1973) as refitted by Dickson and Millero (1987).

### 2.7 Statistical analyses

Relationships among shell parameters (whorls, diameter, amount of shell material, and shell condition via LDX) were examined relative to each other using a simple linear model in the computing language R, version 3.6.0 (R Core Team, 2019) using the RStudio interface (RStudio Team, 2016). To account for running multiple comparisons, *p*-values were corrected using both the more conservative Bonferroni correction, and the less conservative false discovery rate (FDR) (Benjamini and Hochberg, 1995). The $R^2$, Bonferroni-adjusted p-value (*p* Bon.), and FDR-adjusted p-value (*p* FDR) are reported for each comparison in the text and in Table S5.

### 3 Results

Pteropod shell condition varied throughout the course of the experiment, with LDX rankings ranging from 0 (pristine, transparent and lustrous shell) to 4 (shell highly altered, opaque-white and lusterless shell with surface layer dissolution) and shell opacity values ranging between 0.17 (pristine, transparent shell) and 0.74 (highly altered opaque, white shell) (Table S3). Scanning electron microscopy showed the majority of dissolution was concentrated on the outside of the shell up to LDX rankings of 2.5. At higher values, both internal and external walls display evidence of dissolution, and in some cases,

the external surface has dissolved completely revealing the prismatic shell layer (Fig. 2, Fig. S2). The impact of preservation method on pteropod shell condition in this study was determined by comparing the time spent in the sediment trap with the condition of the shells (Fig. 4). Shell condition did not deteriorate with the amount of time spent in the trap (Fig. 4). Although there was a statistically significant relationship ($R^2 = 0.357$, $p$ Bon. $= 5.17*10^{-5}$, $p$ FDR $= 1.29*10^{-5}$) between time in trap and shell condition, the trend suggests shell condition improves with time in the trap (Fig. 4), which is opposite from the expectation that more time in trap would result in more degradation. The least well-preserved specimens came from the September and October 2013 samples (Fig. 5), and had spent a maximum of 2–6 weeks in the sediment trap cup (Fig. 4). The most well-preserved specimens came from June and December 2013 and had spent a maximum of 20–22 weeks in the sediment trap cup (Figs. 4, 5).

The pteropod shells varied in number of whorls, diameter, amount of shell material, and modal shell thickness both within and among samples throughout the year in the Cariaco Basin (Fig. 6, Table S1). The number of whorls varied between 2 1/4 and 2 7/8, and displayed no overall trend through the 11-month study (Fig. 6 a; Table S1). Shell diameter varied in samples collected through the year: specimens from March 2013 had the greatest shell diameters (average 1.70 mm), and shells in the rest of the study period (June 2013 – February 2014) ranged from 0.68 to 1.40 mm in diameter with an average of 0.98 mm (Fig. 6 b). The amount of shell material followed a similar pattern to shell diameter, with specimens from March 2013 containing the greatest amount of shell material (0.104 mm$^3$) and specimens from June 2013 – February 2014 ranging from 0.005 to 0.038 mm$^3$, with an average amount of 0.021 mm$^3$ (Fig. 6 c). The modal thickness of the shells of *Heliconoides inflatus* also varied through the year (Fig. 6 d; Table S1). The thickest shells were sampled in March 2013, with an average modal shell thickness of 0.018 mm, and the thinnest shells were sampled in September 2013, with an average modal shell thickness of 0.009 mm (Fig. 6 d). There was a weak, but statistically significant correlation between shell diameter and the number of whorls which remained when analyzing the subset of complete shells (Table S1) (whole dataset: $R_2 = 0.074$, $p$ Bon. $= 0.415$, $p$ FDR $= 0.057$; subset dataset (Table S1, Fig. S4): $R_2 = 0.101$, $p$ Bon. $= 0.513$, $p$ FDR, $0.057$). As shell diameter, thickness, and amount of shell material are related to size, unsurprisingly, there were significant correlations between shell diameter and amount of shell material ($R^2 = 0.819$, $p$ Bon. $= 2.20*10^{-15}$, $p$ FDR $= 2.20*10^{-15}$), shell diameter and shell thickness ($R^2 = 0.582$, $p$ Bon. $= 1.79*10^{-8}$, $p$ FDR $= 5.97*10^{-9}$), and shell thickness and amount of shell material ($R^2 = 0.680$, $p$ Bon. $= 6.09*10^{-11}$, $p$ FDR $= 3.05*10^{-11}$). These results highlight that larger shells are generally thicker and contain more shell material.

The modal shell thickness of the specimens, used in this study as a calcification metric, was analyzed with respect to the water column properties in the Cariaco Basin (Fig. 7; Table S4). Water chemistry measurements from 55 m depth were used because this was the closest water sample to the most recent of *H. inflatus* calcification depth estimate of 75 m (Keul et al., 2017). The Cariaco Basin was supersaturated with respect to aragonite throughout the studied interval ($\Omega_{arag}$ range 2.28 – 3.59), and the thickest shells formed when the aragonite saturation was the lowest (March 2013, Dec 2013 – Feb 2014;

average $\Omega_{arag}$ 2.49) (Fig. 7 c; Table S4). Specimens collected during the upwelling season (December – April) were compared to those from the rainy season when there was no upwelling (August – November), using a Welch's t-test, because the two groups had different variances and unequal sample sizes, prohibiting the use of a Student's *t*-test (Revelle, 2018). Pteropod shells were 40% thicker during the upwelling season, when water temperatures were lower and nutrient concentrations were higher, than during the rainy season when oligotrophic conditions prevailed (Welch's *t*-test: $p = 4.41$ x $10^{-4}$; Table S6; Figs. 7, 8). Pteropod shell diameters were also 20% larger during the upwelling season than during the rainy season (Welch's *t*-test: $p = 0.0080$; Table S6).

Because shell diameter and shell thickness are related to the overall size of a specimen, the influence of shell diameter on shell thickness was removed using a simple linear regression model of thickness as a function of diameter. Analysis of the residuals of this model, hereafter referred to as 'residual thickness', found that specimens sampled during the upwelling season had significantly higher residual thicknesses than those sampled during the rainy season (Welch's *t*-test: $p = 0.0260$; Figure S5; Table S6), indicating that water column properties impact calcification regardless of shell size.

## 4 Discussion

### 4.1 Shell condition

This study focuses on how the interplay of biotic and abiotic factors impact the shell characteristics of the pteropod *Heliconoides inflatus* in the Cariaco Basin. The specimens used in this study were collected using a sediment trap, adding a third variable, taphonomy. Pteropod shell condition was assessed using both the LDX (Gerhardt et al., 2000; Gerhardt and Henrich, 2001) and opacity (Bergan et al., 2017) methods. By comparing the results from these two methods, we found that the opacity scale lacked sensitivity to changes in shell condition at LDX values of 2 (opaque white shells with lustrous surface) and higher (Fig. 9). When pteropod shells dissolve, the shell transparency changes first, from transparent, to milky-white, to opaque-white, followed by the surface texture (Gerhardt and Henrich, 2001). Because the opacity method is based on greyscale values of light microscope images, it quantifies the change in shell color but not texture, meaning this method is only sensitive to shell condition changes in the early stages of dissolution (LDX stages 0 – 2; Fig. 9). Since the opacity method was designed to assess pteropods from an incubation experiment, it was intended to capture the earliest stages of dissolution (Bergan et al., 2017). Because of the wide range of shell conditions of the specimens in this study, spanning both changes in color and texture, all shell condition analyses are based on LDX measurements.

The shell condition of specimens from sediment trap samples has the potential to be altered via three mechanisms: 1) dissolution in the water column when the organism is alive; 2) dissolution in the water column when the organism is dead; 3) alteration in the sediment trap cup associated with the preservative. The water in the Cariaco Basin was supersaturated with respect to aragonite throughout the study. The thin, aragonite shells of the pteropods would therefore have been chemically

stable in the water column and thus it is unlikely that they underwent in-life dissolution. Furthermore, there is no evidence of patchy dissolution in pristine shells, or those which have undergone dissolution (Fig. 2, Figs. S1, S2), such as has been observed in pteropod shells undergoing in-life dissolution in naturally undersaturated environments (Peck et al., 2016; 2018).

Once a pteropod dies, the degradation of the organic body and associated acid production has been found to cause significant dissolution on the internal walls of the pteropod shell, even in an aragonite-supersaturated water column (Oakes et al., 2019a). Dissolution can occur on the outside of the shells from the breakdown of free-floating organic matter in the water column creating aragonite-undersaturated microenvironments in an otherwise aragonite-supersaturated water column (Milliman, 1999). LDX rankings show the greatest amount of shell alteration occurred in specimens from the September –
October 2013 samples (Fig. 5). Scanning Electron Microscopy reveals that the majority of this dissolution occurred on the outside of the shells (Fig. 2, Fig. S2). During September and October, water temperatures at 55 m were at their highest (Fig. 7 b). These warm temperatures would have increased the rate of microbial breakdown of both the organic body within the shell (Oakes et al., 2019a), and in the free-floating decaying organic matter in the water column (Lohmann, 1995; Milliman et al., 1999; Schiebel et al., 2007). The shells of the organisms that died during the warmer months likely encountered more
aragonite-undersaturated microenvironments associated with this organic matter breakdown as they fell through the water column and into the trap, increasing the rates of dissolution of these shells relative to those trapped during cooler months. These results could have been further complicated by the presence of swimmers, which would have entered the trap live and therefore would not have undergone any dissolution in the water column. If there was an increase in swimmers entering the traps at one time of year relative to another, it could be interpreted as less water column breakdown during these months.
The most pristine shells in this study entered the trap in June and December, suggesting that there was not a seasonal pattern to swimmer frequency. We therefore assume that the number of swimmers entering the sediment trap is constant throughout the year and therefore does not affect the seasonal trends reported above.

The borate-buffered formalin solution used to preserve sediment trap samples has been shown to influence the condition of
pteropod shells (Oakes et al., 2019b). We found that shell condition did not deteriorate with time spent in the sediment trap cups (Fig. 4). Preservation-associated dissolution would have affected both the internal and external walls of the shell. SEM images reveal that the internal shell walls were only impacted by dissolution at LDX values of 2.5 and higher (Fig. 2 d, h, l), indicating that the preservative did not cause dissolution. Specimens that had undergone the most dissolution were sampled during the warmest months, which happened to coincide with the shortest amount of time in the trap. This produced an
apparent trend of improving shell condition with time in the trap (Figs. 4, 5). This suggests that the preservative in the sediment trap collection cups effectively minimized post-collection sample degradation and that any sediment trap-associated changes in shell condition likely happened on timescales of 2 weeks or less, the amount of time the specimens were in the final sediment trap collection cup before trap recovery.

## 4.2 Pteropod development

Assessing the number of whorls, shell diameter, amount of shell material, and shell thickness provides an integrated view of *H. inflatus* shell growth in the Cariaco Basin. The number of whorls varies both within and among samples throughout the year. Although there is a weak relationship between the number of whorls and shell diameter (Fig. 6 a, b), *H. inflatus* displays considerable plasticity during growth. These measurements support the observations of Janssen (1990) who found that both the number and diameter of the whorls of *H. inflatus* increase irregularly. There are no patterns in the overall trend of *H. inflatus* shell diameter through the year in the Cariaco Basin (Fig. 6 b), which suggests there are no cohorts. Another low latitude study found that *H. inflatus* collected off the coast of Barbados reproduced throughout the year (Wells, 1976a), although *H. inflatus* from off Bermuda in the Sargasso Sea have been shown to spawn in the spring (Almogi-Labin et al., 1988). Both Van der Spoel (1967) and Janssen (2004) have described variability in the shape and position of the aperture tooth in *H. inflatus*, which could be attributed to intraspecific or interspecific variations. As there has not been any genetic work conducted on *H. inflatus* from the Caribbean, we cannot be sure that the variability we see in shell shape cannot be attributed to two or more genetically-defined species.

## 4.3 Pteropod growth and water column properties

Because of their shell chemistry, pteropods have been proposed as biological indicators of aragonite saturation (Bednaršek et al., 2017, 2019). In this study we used shell thickness as a metric of calcification. In the Cariaco Basin, the water is permanently supersaturated with respect to aragonite (i.e., $\Omega_{arag} > 1$). In this aragonite-supersaturated setting, the thickness of pteropod shells does not correlate with aragonite saturation, and the thinnest shells were found when the aragonite saturation was the highest (Aug – Nov 2013 – average $\Omega_{arag}$ 3.26) (Fig. 7 c). Instead, the shell thickness of *H. inflatus* varies with the physical oceanographic conditions in the Cariaco Basin, with median shell thickness increasing by 40% during times of upwelling (Fig. 8), when nutrient rich waters are brought to the surface, relative to shells forming during the rainy season when there is no-upwelling and oligotrophic conditions prevail (Figs. 7, 8; Table S6) (Muller-Karger et al., 2001, 2019). These upwelling-related nutrient changes in the Cariaco Basin have been shown to correspond with increases in organic carbon flux and diatom blooms (Thunell et al., 2000; Romero et al., 2009), indicating that pteropod food supply (Lalli and Gilmer, 1989) increases during upwelling conditions. The diameters of pteropod shells sampled during times of upwelling were 20% larger than those formed during the rainy season (Table S6), and the trend of increased shell thickness during times of upwelling still holds once the influence of shell diameter on shell thickness is removed (Fig. S5, Table S6). The observed changes in *H. inflatus* modal shell thickness and diameter are therefore likely linked to changes in nutrients, and therefore food supply, in the Cariaco Basin through the year.

The link between food availability and shell growth has been proposed for another species of pteropod in the same family as *H. inflatus*, *Limacina retroversa*, which was found to form smaller shells when food resources were limited (Meinecke and Wefer, 1990). Furthermore, the availability of food has been found to offset, or even negate, the negative effects of increased

pCO$_2$ levels or low pH in other groups of marine calcifiers such as mussels, oysters, and corals (Heinemann et al., 2012; Hettinger et al., 2013; Kroeker et al., 2016; Ramajo et al., 2016; Thomsen et al., 2013; Towle et al., 2015), presumably because organisms require energy for biomineralization (Palmer, 1992). Feeding rates in calcifiers can also be affected by acidified conditions. The effects vary according to phylum, feeding style, life stage, and exposure time, with the feeding
rates of suspension-feeding molluscs particularly susceptible to decrease with increased CO$_2$ (Clements and Darrow, 2018). There have not been any studies conducted on the response of pteropods to varying acidification and food availability conditions, however, we assume that as in other groups of marine calcifiers, food availability plays an important role in calcification. This body of research supports the inference made from the findings of this study that when seawater is supersaturated with respect to aragonite, such as in the Cariaco Basin, food availability is the main control of *H. inflatus*
shell growth.

**4.4 Further work**

Micro-CT scanning enables pteropod shells to be digitized in three dimensions, creating the opportunity for more complex quantitative analyses of shell shape and parameters than presented in this study. Despite their geometrically simple shapes,
gastropod shells are particularly challenging to perform geometric morphometric analyses on because of their lack of fixed landmark points (Liew et al., 2016). There has been recent progress in the field of gastropod 3D geometric morphometrics, to understand variability in shell form (Liew et al., 2016) and changes in shell calcification associated with ocean acidification (Harvey et al., 2018). These analyses are beyond the scope of this study; however, the CT data are available on Morphosource and therefore can be used for morphometric analyses.

**5 Conclusions**

In this study, we analyzed the shell diameter, number of whorls, thickness, amount of shell material, and shell condition of *Heliconoides inflatus*, a species of pteropod from the Cariaco Basin, over an 11-month period. Because specimens in this study came from a sediment trap, the impact of time in the sediment trap on shell condition was analyzed. Shells were
assessed using both the LDX and opacity methods, however, as the opacity method was only sensitive to changes in shell condition at LDX scores of two or lower, and therefore LDX was used for all analyses. Although all shells had undergone some alteration, shell condition did not deteriorate with increased time in the sediment trap cup. The most poorly preserved specimens came from sediment trap samples collected when seawater temperatures were the highest, suggesting that dead specimens were affected by dissolution from increased rates of microbial breakdown of organic matter both in the water
column, and within the pteropod shell.

The size, number of whorls, thickness and amount of shell material in the shells of *H. inflatus* vary throughout the year, and therefore are likely to be influenced by external factors. Water chemistry in the Cariaco Basin is controlled by the movement of the ITCZ and has two distinct phases: an upwelling phase and a non-upwelling, oligotrophic phase. We find that *H.*

*inflatus* produces larger, thicker shells during times of upwelling, when food availability is greater. The Cariaco Basin was supersaturated with respect to aragonite throughout the study period (i.e. $\Omega_{arag} > 1$) and shell thickness does not correlate with $\Omega_{arag}$. This demonstrates that in this aragonite-supersaturated setting, the availability of food has a greater control on shell formation than aragonite saturation. This pattern has been seen in other groups of molluscs, such as oysters and mussels and underlines the necessity of assessing pteropod shell parameters and dissolution in the context of multiple biotic and abiotic factors, not just aragonite-saturation. We hope that the baseline dataset of pteropod shell parameters presented in this study is the first of many focused regional studies around the world. These datasets will enable the quantification of the response of this sentinel group to ocean acidification.

## 6 Data availability

The data which support the conclusions in this manuscript are available in the tables, figures, references, and supplemental materials. CT data will be made available on MorphoSource (https://www.morphosource.org) once the manuscript is accepted.

## 7 Sample availability

Specimens have been deposited in the Malacology collection at the Academy of Natural Sciences of Drexel University, Philadelphia, PA, USA (ANSP). A sample list, including the ANSP catalog numbers, can be found in Table 1.

## 8 Author contributions

Following CRediT: Conceptualization (RLO), Data curation (RLO, JAS, PC), Formal analysis (RLO), Funding acquisition (RLO, JAS, TJB), Investigation (RLO), Methodology (RLO), Project administration (RLO), Resources (BM, RT, CD – University of South Carolina, JU, WY, MH), Software (RLO), Supervision (RLO, JAS, TJB), Validation (RLO, JAS), Visualization (RLO), Writing – original draft (RLO), Writing – reviewing and editing (RLO, JAS)

## 9 Competing interests

This is an original submission and the authors do not declare any conflicts of interest.

## 10 Acknowledgements

The authors would like to thank B. Marshall for picking the first batch of specimens whilst in the midst of finishing her Ph.D, and C. Davis for picking the second set of samples, and for helpful discussion about sample processing. Light

microscopy was performed in the Paleobotany Lab at the Pennsylvania State University thanks to P. Wilf, and at the Academy of Natural Sciences thanks to R. Thomas and C. Vito. Scanning electron microscopy was performed at the Nanoscale Characterization Facility at the Singh Center for Nanotechnology at the University of Pennsylvania, Philadelphia thanks to J. Ford. CT scanning was performed at GE in Lewistown thanks to J. Urbanksi and W. Yetter, and the American

Museum of Natural History in New York thanks to M. Hill and M. Siddall. Thanks to T. Woodger and J. Foster for logistical support. We thank E. Tappa for information about the CARIACO sediment trap and for providing a map, and T. Bralower, M. Potapova and G. Rosenberg for discussions that helped to shape this manuscript. Thanks to K. Kimoto, A. Almogi-Labin, and an anonymous reviewer for their thoughtful, constructive comments which helped to improve this manuscript.     . This work was funded by the Deike Research Grant awarded to T. Bralower and R. Oakes, and R. Oakes was supported by the

John J. & Anna H. Gallagher Fellowship, The Academy of Natural Sciences of Drexel University.

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

**12 Figures**

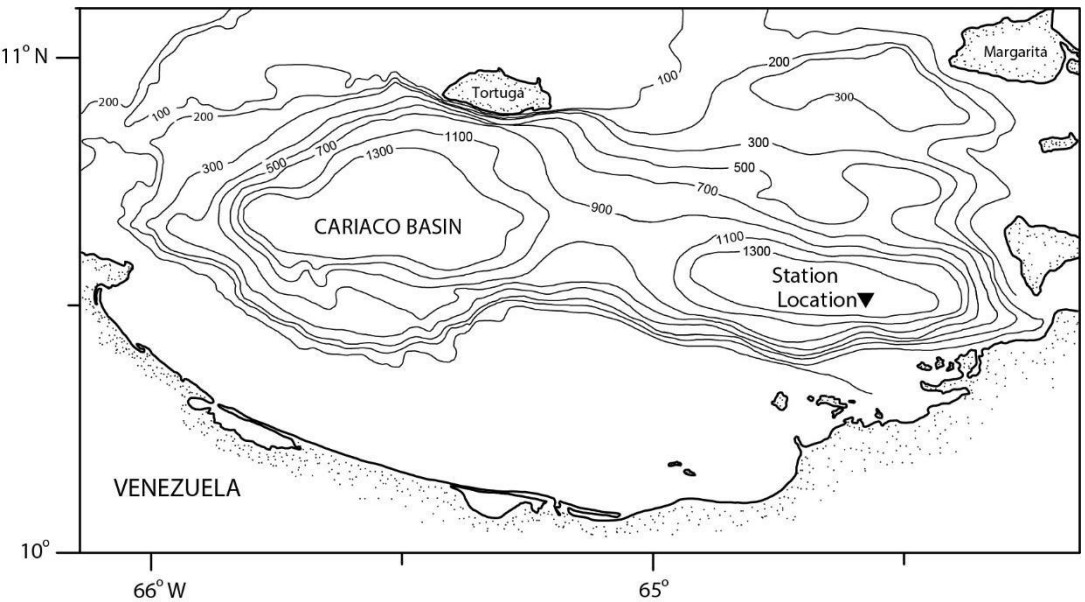

**Figure 1: Bathymetric map of the Cariaco Basin. The location of the sediment trap (10° 30.0' N, 64° 38.5' W) is marked with a triangle (modified from Marshall et al., 2013).**

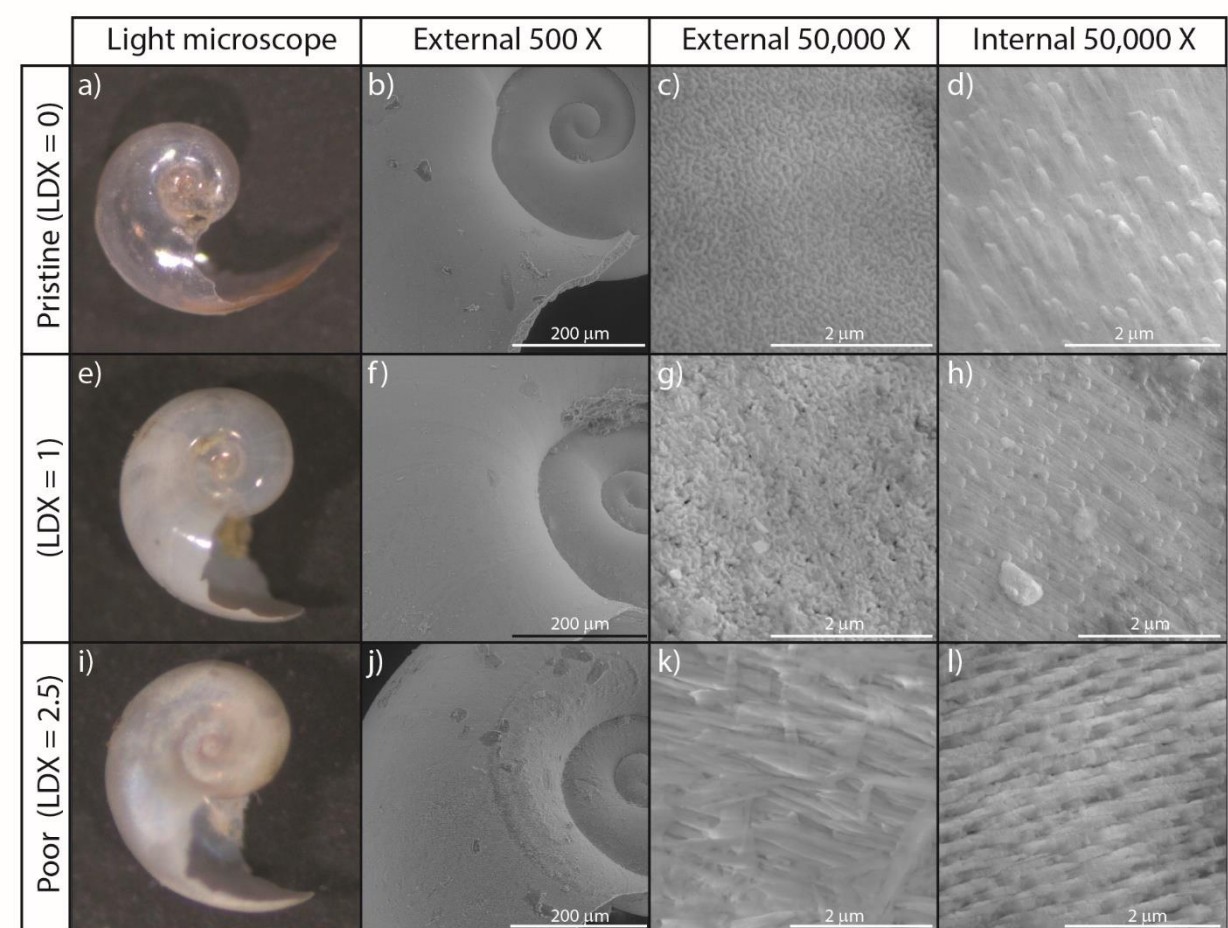

|  | Light microscope | External 500 X | External 50,000 X | Internal 50,000 X |

Figure 2: Light microscope (a, e, i), and scanning electron microscope images of the external (b, c, f, g, j, k), and internal (d, h, l) faces of *H. inflatus* shells from the Cariaco Basin. Light microscope images show the shell changing from pristine and glassy to opaque and white with increasing dissolution. This change is accompanied by an increase in pocking on the shell surface to reveal the tops of the prismatic crystals (c, g) and then the whole prismatic layer (k). The topography on the internal face is due to the terminations of the cross-lamellar crystals intersecting with the internal face (d, h). These become more distinct as dissolution increases the porosity of the internal face (l).

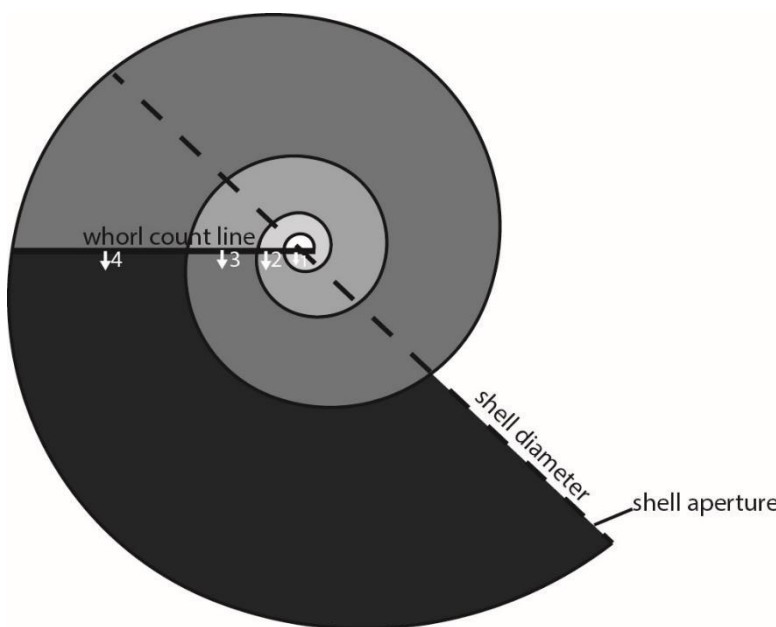

**Figure 3: Schematic diagram of a pteropod shell demonstrating how shell diameter (the metric used for size) was measured, and how the number of whorls was counted. Following the methods outlined in Janssen (2007), a straight line is drawn across the shell separating the semi-circular nucleus (center) from the rest of the shell. Whorls are then counted as 360 ° rotation from the straight line, marked in progressively darker shades of grey, until the aperture of the shell is reached. The number of whorls is recorded with an accuracy of an eighth of a whorl. The shell in the schematic diagram has 3 3/8 whorls.**

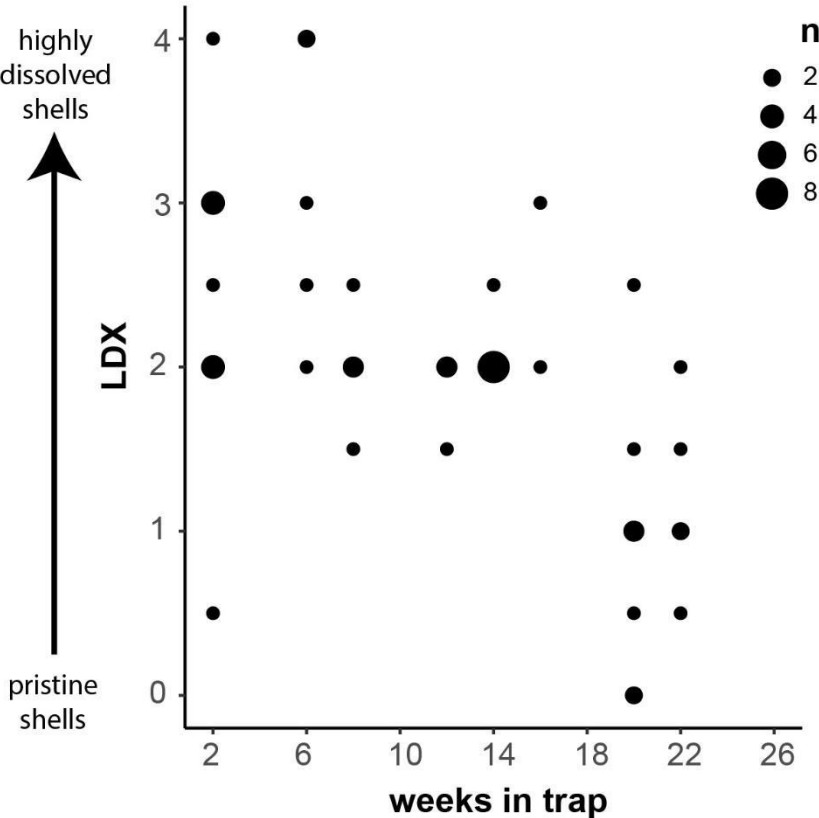

**Figure 4: Shell condition of *Heliconoides inflatus*, ranked on the *Limacina* Dissolution Index (LDX) scale, plotted against the maximum amount of time specimens spent in the sediment trap (i.e., the number of weeks from the trap opening time). The size of the symbols corresponds to n, the number of specimens plotted at a given point.**

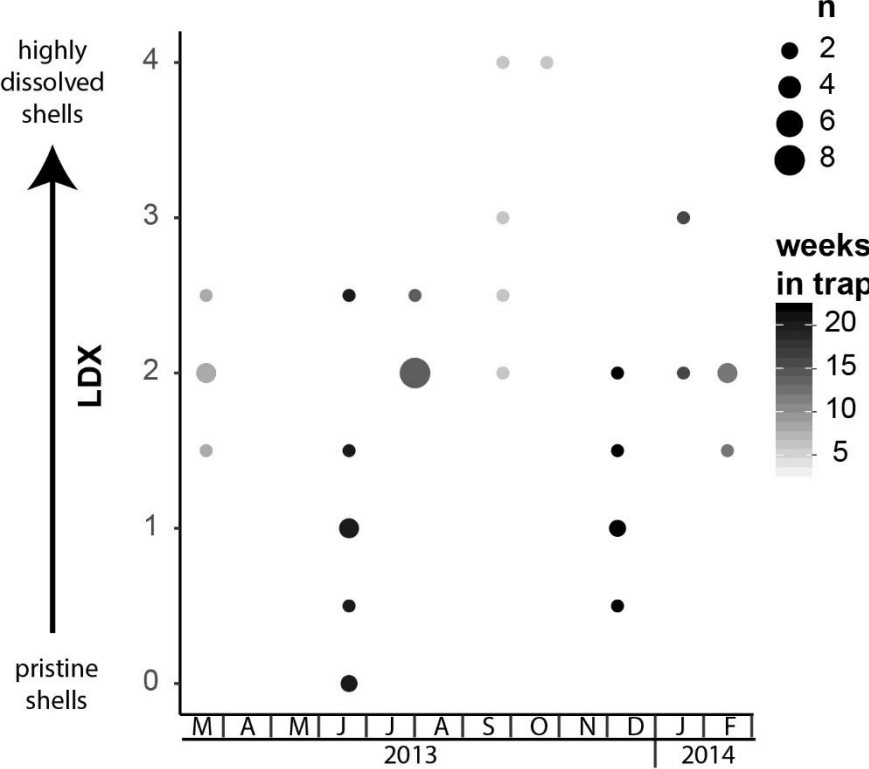

**Figure 5: Shell condition of *Heliconoides inflatus*, ranked using the *Limacina* Dissolution Index (LDX) scale, over the study period. The samples with the poorest preservation are from September and October 2013 when water temperatures were the highest. The size of the circles corresponds to n, the number of specimens plotted at a given point, and the color of the circles corresponds to the maximum number of weeks specimens were in the trap.**

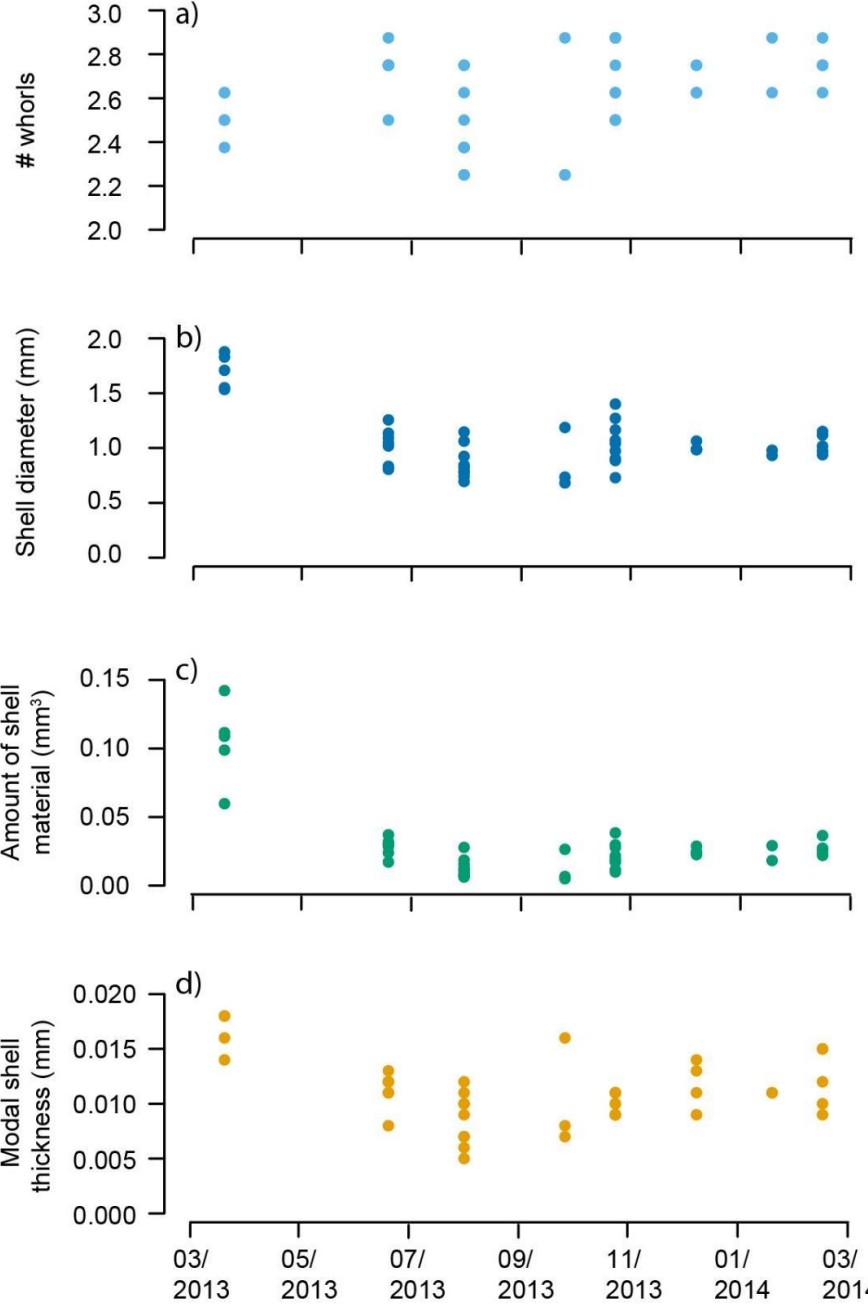

**Figure 6:** *Heliconoides inflatus*: a) number of whorls; b) shell diameter; c) amount of shell material; and d) modal shell thickness throughout the year in the Cariaco Basin. Each point represents an individual specimen.

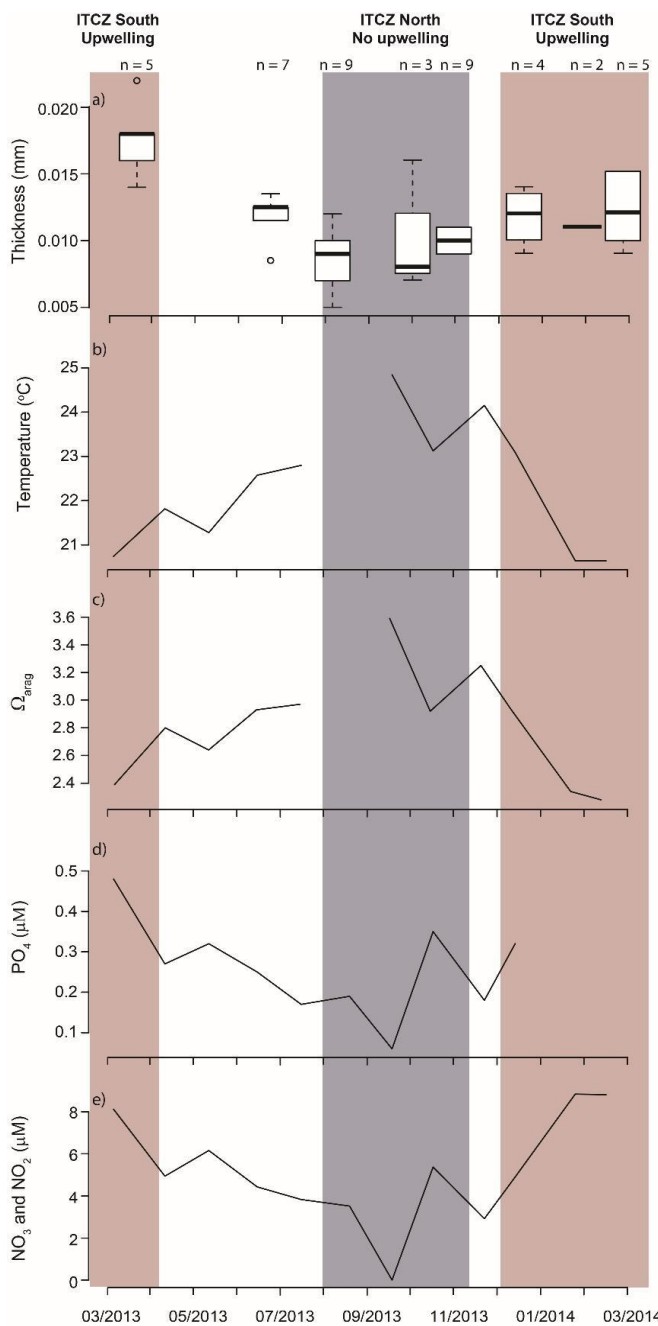

**Figure 7: Shell thickness and water column properties plotted over the study period: a) Heliconoides inflatus modal shell thickness, b) seawater temperature, c) $\Omega_{arag}$, d) PO$_4$, and e) NO$_2$ and NO$_3$. Nutrient concentrations (d and e) are plotted as proxies for upwelling and food availability (Romero et al., 2009; Thunell et al., 2000). All water column measurements (b-e) are from 55 m depth because this is the water sample closest to the predicted calcification depth of *Heliconoides inflatus* (Keul et al., 2017). The upwelling season is indicated by a red box, and the rainy season, when there is no upwelling, is indicated by a grey box.**

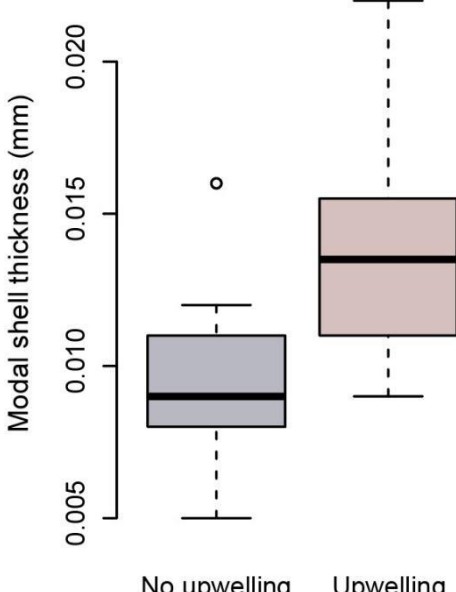

**Figure 8: Modal shell thicknesses of specimens from times of upwelling (red) and times of no upwelling (grey) in the Cariaco Basin. Specimens collected during times of upwelling are significantly thicker than those which formed at times with no upwelling (Welch's t-test: $p$ = 4.4 x 10$^{-4}$).**

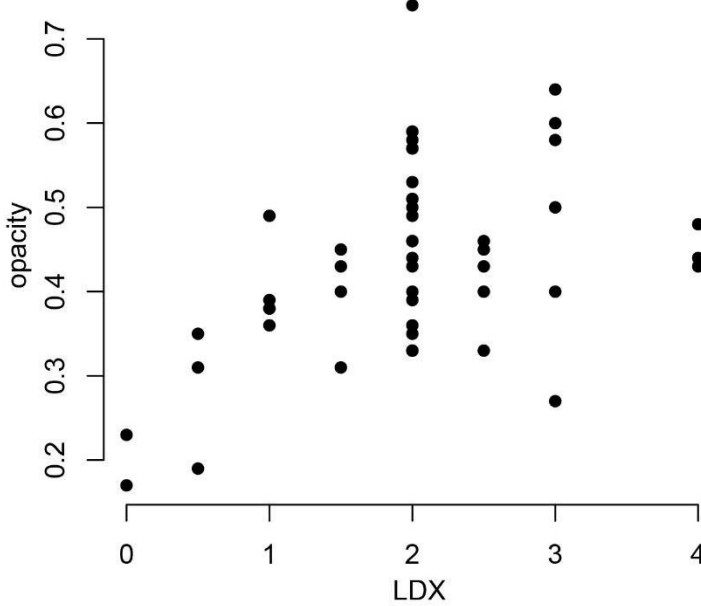

**Figure 9: Shell condition of *Heliconoides inflatus*, ranked on the *Limacina* Dissolution Index (LDX), plotted against shell condition of the same shells quantified using the opacity scale. LDX and opacity are positively correlated until LDX scores of 2, at which point there is no correlation between LDX and opacity. This breakdown is likely due to the changes in surface texture of the pteropod shell from shiny to matte. The texture change linked to dissolution is a factor when assigning values on the LDX scale but**
10    **as the color and opacity do not change, it is not detected by the opacity scale.**

## 13 Tables

| Sample | Trap date | Light microscope imaged | CT scanned | ANSP catalogue no. |
|---|---|---|---|---|
| CAR34Z#10 | 21/03/2013 | 5 | 5 | 477912 |
| CAR35Z#04 | 20/06/2013 | 8 | 7 | 477913 |
| CAR35Z#07 | 01/08/2013 | 9 | 9 | 477914 |
| CAR35Z#11 | 26/09/2013 | 5 | 3 | 477915 |
| CAR35Z#13 | 24/10/2013 | 11 | 9 | 477916 |
| CAR36Z#03 | 08/12/2013 | 5 | 4 | 477917 |
| CAR36#06 | 19/01/2014 | 2 | 2 | 477918 |
| CAR36#08 | 16/02/2014 | 4 | 5 | 477919 |
|  |  | **49** | **44** |  |

**Table 1: Number of specimens imaged and CT scanned from each sediment trap cup.**

