# Peer review of "Determining how biotic and abiotic variables affect the shell condition and parameters of *Heliconoides inflatus* pteropods in the Cariaco Basin"

_Biogeosciences, 2019_

## Referee Comment (RC1) · Anonymous Referee #1 · 29 Oct 2019

**GENERAL COMMENTS**

The aim of this study is to describe annual variability of different shell growth parameters (thickness, diameter, number of whorls, amount of shell material) of the pteropod Heliconoides inflatus in the Cariaco Basin (Venezuelan Shelf). Additionally shell condition was analyzed applying the Limacina Dissolution Index (LDX). Pteropod samples were collected over a year period from a sediment trap and compared to prevailing carbonate chemistry and nutrient conditions with the goal to entangle driving abiotic or respectively biotic factors of the various measures. The authors found that food avail-
ability has a greater control on shell formation than aragonite saturation and that shell condition was not altered with time spent in the sediment trap cup. Hence, the results can serve as baseline data to better quantify the response of this highly vulnerable organism group to ocean acidification (OA) by disentangling abiotic from biotic factors that impact on shell formation.

I think this study is very interesting and addressing a very important question in relation with consequences of OA on highly vulnerable thecosome pteropods. It gives strong in situ evidence that food availability and energy constraints have a major potential to mitigate abiotic stress and shows nicely that various shell parameters indicative for growth and calcification did not depend on the saturation state of aragonite, at least not in the range observed (always above 2).

From my understanding, the purpose of the study was twofold: 1) How does length of time (preservative) in the trap impact shell condition and potentially lead to false conclusions in the OA context? 2) Do changes in water column properties affect shells and how or which? Hence, point 1 looks at dead organisms, point 2 affects live organisms in the water column (including the carbonate chemistry history pteropods experienced in the past). In this context, my main criticism is that the author did not distinguish between processes that happened when pteropods were still alive (in the water column) and already dead (in the water column and the sediment) particularly with respect to potential shell degradation they observed on the preserved samples. Did the authors simply assume, that shell integrity was intact as long as organisms dwelled in the water column alive? Might indeed be reasonable to assume but the authors need to state clearly in their ms what their opinion on that is and whether/when they talk about live or dead organism. Furthermore, one important issue with sediment trap samples is that pteropods might have entered them as "swimmers" not as dead individuals that simply sank into the trap. This problem should be mentioned in the introduction and picked up later in the discussion again, would that impact the conclusions to draw from the results?
**TITLE**

The title does not reflect the study content well enough. For example micro-CT is not even mentioned in the abstract and shell thickness is only one out of a set of measured parameters mentioned in the abstract. LDX is much more prominent in the abstract instead. Also, I think the title should reflect that the ms is about sediment trap samples of H. inflatus. Please change title accordingly to maybe something like this: "Assessing abiotic and biotic impact on annual variability of shell condition of the pteropod Heliconoides inflatus in the Cariaco Basin: shell dissolution index, size and thickness as revealed from sediment trap samples."

**ABSTRACT**

L4–6: This study does not deal with natural variability of pteropods (in terms of abundance of which "variability" is usually understood if not stated otherwise), neither is it discussed. Either remove this sentence or rephrase to harmonize with the variability you are actually focusing on (shell growths parameters).

L11: remove "with"

L14/15: Are the authors talking about dead or live individuals?

L19: ... in shell characteristics of H. inflatus of trapped pteropods...

**INTRODUCTION**

Section 1.2: The authors should shortly mention the problem of collecting live pteropods ("swimmers") in sediment traps and how that could have affected their work approach and results. (Alternatively it might be mentioned on P4 last paragraph). Throughout the ms, they need to make clear whether they talk about live or dead organisms.

P4L6: Lischka and Riebesell 2017 (Polar Biol, Volume 40) also studied metabolic re-

BGD
sponse of pteropods (oxygen consumption).

P4L24: through misses the "r"

P5L33: remove comma between body and whorl

MATERIAL & METHODS:

P6L10: Please detail at what temperature and for how long shells were dried.

DISCUSSION

P10L31/32: Could any changes detected originate in the time prior collection in the trap during live in the water column?

P11L7/8: How can the authors know, pteropods were dead already? How likely is it that shell deterioration happened on the live organism? The assumption that any shell degradation took place only when organisms were dead already, is this simply based on the assumption that under aragonite supersaturated conditions no shell deterioration happened? If so, state clearly and support your view.

P11L20: ... in the overall trend... (remove "is no")

FIGURES

Fig. 4: It would help the clarity of the figure if September, June, December (mentioned in the text) could be indicated on the x-axis.

Fig. 8: Italics for Heliconoides inflatus

---

## Referee Comment (RC2) · K. Kimoto (Referee) · 5 Dec 2019

kimopy@jamstec.go.jp Received and published: 5 December 2019

This manuscript is described that biological responses Below are the comments:

This manuscript describes the biometrics of pteropod shell and its degradation based on sediment trap samples in the Cariaco Basin. This kind of works of pteropod shells using the sediment trap samples are insufficient, so it is very important to trace the biological responses to the ocean acidification and related ocean environmental changes. Especially the information of tropical species is less. In this sense, this work has the

potential to become the base and develop criteria for this kind of study. Below I pointed out some concerning issues for this ms and make the comments

1. The relationship between shell length and whorls. According to photos in supplemental document, the aperture of some shells was damaged or showed bad preservation. The author inferred that shell diameter and whorl does not show clear relationship, but if a part of aperture had lost by dissolution and/or physical damage, its relationship between shell length and whorl might become uncertainty. If the plankton tow samples are available, the authors should use those plankton samples, not sediment trap ones. Or at all possible, the authors should use the only perfect shell in order to interpret length-whorl relationship.

2. Shell dissolution: How and when? 2. shell dissolution: How and when The authors described that preservation states of shells in the sediment trap was not related to the duration time, so it might be negligible dissolution in the sediment trap collection cups. If this is correct, shell dissolution occurred at the water column, and it was associated with microzoo/bacterial activity which was decomposing organic tissues. My questions are that 1) in this case, does shell dissolution occurred at the inside, and outside of shell is sufficiently preserved? Can the authors show this evidence? Based on the photographs on the supplement material, surface texture of some shells looks like cloudy and lost their luster, indicating dissolution of outer shell. I am wondering that the decision of less dissolution in the sediment trap collection cups based on the result of relationship between residence time and LDX might be insufficient. In other words, I infer that shell dissolution occurs not only by microorganisms/bacterial activity but also postdepositional oxidization in the sediment trap cups, as authors mentioned. I am understanding that this certification is very difficult, but the authors are using SEM. so please show some possibilities from the direct observations of materials. Another possibility, is it available the comparison between organic carbon of the samples and LDX ? Highly input of organic carbon flux induce carbonate dissolution at the inside of sediment trap cup. 2) Relating to above, I understood this study is the first, and

make the baseline of this kind of pteropod study, but it is bit unclear the main subject and purposes. If the authors interpretation is correct, does the pteropod shell of this species /or in this region not become an index of ocean acidification? I suppose that the authors want to make the criteria as OA index by using pteropod shell, but in this case, I think that shell preservation states indicate microorganisms activity in the pteropod shell.

3. I think it is very important finding that shell thickness does not have relationship with surrounding omega value (but still supersaturated). I strongly agree with the authors that they have resistance characteristics to small changes of saturation states and depends on available food to build their shells. However, the authors did not show what kind of food is important for their prey. If their main food is phytoplankton, please show their annual variations through a year instead of nutrient concentrations (Or is it possible to show the number of diatom bulbs in the sediment trap cups?) Because their food is particulate matters, not chemical component. It might be a good evidence to indicate their food availability.

4. The authors did not touch the phylogenetic variation of the species, but I am wondering the possibility of mixture of some lineages of this species. H. inflatus is certificated as a single-genetic species around the Cariaco basin? Or exists some cryptic species? If the author has this kind of information, please mention it for just confirmation. It is possible that the plasticity of shell (shell length, number of whorls) of this species that author mentioned is related with the phylogenetic variations.

5. Can the author interpret about morphological implication from the microXCT analysis? Because it is very powerful tool and shows huge possibility for morphological information. If the authors want to indicate some suggestive issues, please make comment for following researchers and future study

---

## Author Comment (AC1) · 22 Jan 2020

Reviewer comments have been copied below and author responses are listed underneath them. Page and line numbers refer to the tracked changes version of the manuscript.

Reviewer # 1 - General Comments: The aim of this study is to describe annual variability of different shell growth parameters (thickness, diameter, number of whorls, amount of shell material) of the pteropod Heliconoides inflatus in the Cariaco Basin (Venezuelan Shelf). Additionally, shell condition was analyzed applying the Limacina Dissolution Index (LDX). Pteropod samples were collected over a year period from a sediment trap and compared to prevailing carbonate chemistry and nutrient conditions with the goal to entangle driving abiotic or respectively biotic factors of the various measures. The authors found that food availability has a greater control on shell formation than aragonite saturation and that shell condition was not altered with time spent in the sediment trap cup. Hence, the results can serve as baseline data to better quantify the response of this highly vulnerable organism group to ocean acidification (OA) by disentangling abiotic from biotic factors that impact on shell formation.

I think this study is very interesting and addressing a very important question in relation with consequences of OA on highly vulnerable thecosome pteropods. It gives strong in situ evidence that food availability and energy constraints have a major potential to mitigate abiotic stress and shows nicely that various shell parameters indicative for growth and calcification did not depend on the saturation state of aragonite, at least not in the range observed (always above 2).

From my understanding, the purpose of the study was twofold: 1) How does length of time (preservative) in the trap impact shell condition and potentially lead to false conclusions in the OA context? 2) Do changes in water column properties affect shells and how or which? Hence, point 1 looks at dead organisms, point 2 affects live organisms in the water column (including the carbonate chemistry history pteropods experienced in the past). In this context, my main criticism is that the author did not distinguish between processes that happened when pteropods were still alive (in the water column) and already dead (in the water column and the sediment) particularly with respect to potential shell degradation they observed on the preserved samples. Did the authors simply assume that shell integrity was intact as long as organisms dwelled in the water column alive? Might indeed be reasonable to assume but the authors need to state clearly in their ms what their opinion on that is and whether/when they talk about live or dead organism. Furthermore, one important issue with sediment trap samples is that

pteropods might have entered them as "swimmers" not as dead individuals that simply sank into the trap. This problem should be mentioned in the introduction and picked up later in the discussion again, would that impact the conclusions to draw from the Results?

Author response: We agree with the general comment about the need to discuss processes affecting pteropod shells when the organism is live, dead, and in the preservative. We have added information to the abstract and introduction, and have reformatted our discussion to better outline these processes and the likelihood of shell alteration occurring at each of these stages. Abstract: Pg. 2, lines 10 – 13, Introduction: Pg. 4, line 33 – Pg. 5, line 11, Discussion: Pg. 12, lines 5 – 24). We have also discussed the potential impact of swimmers on the samples in the introduction (Pg. 5, lines 1 – 4) and discussion (Pg. 12, lines 24 – 30) sections.

Reviewer #1 - Specific Comments

Reviewer comment: The title does not reflect the study content well enough. For example micro-CT is not even mentioned in the abstract and shell thickness is only one out of a set of measured parameters mentioned in the abstract. LDX is much more prominent in the abstract instead. Also, I think the title should reflect that the ms is about sediment trap samples of H. inflatus. Please change title accordingly to maybe something like this: "Assessing abiotic and biotic impact on annual variability of shell condition of the pteropod Heliconoides inflatus in the Cariaco Basin: shell dissolution index, size and thickness as revealed from sediment trap samples."

Author response: Thank you for pointing that out. We have changed our title to, "Determining how biotic and abiotic variables affect the shell condition and parameters of Heliconoides inflatus pteropods in the Cariaco Basin", based on your recommendation.

Reviewer comment: L4–6: This study does not deal with natural variability of pteropods (in terms of abundance of which "variability" is usually understood if not stated otherwise), neither is it discussed. Either remove this sentence or rephrase to harmonize

with the variability you are actually focusing on (shell growths parameters).

Author response: We agree that the use of 'variability' could have been interpreted in multiple ways throughout the text. We have changed the text in the abstract to clarify our focus on shell growth parameters. It now reads, "... the biotic and abiotic factors influencing their shell formation and dissolution in the modern ocean need to be quantified and understood." (Pg. 2, lines 5 – 7). We have either changed the wording, or expanded on the meaning of our use of the word 'variability' through the manuscript.

Reviewer comment: L11: remove "with"

Author response: done

Reviewer comment: L14/15: Are the authors talking about dead or live individuals?

Author response: We agree that the differentiation between processes that affected live and dead individuals was unclear. We have restructured the abstract to clarify the mechanisms that can alter shell condition when the animals are live, dead, and in the preservative. "The shell condition of pteropods from sediment traps have the potential to be altered at three stages: 1) when the organisms are live in the water column associated with ocean acidification, 2) when organisms are dead in the water column associated with biotic decay of organic matter, 3) and when organisms are in the sediment trap cup associated with the abiotic alteration by the preservation solution." (Pg. 2, lines 10 – 13).

L19: : : : in shell characteristics of H. inflatus of trapped pteropods: : :

Author response: We acknowledge the importance of conveying the fact these samples are from a sediment trap in the abstract. We have added a section on mechanisms affecting shell condition in specimens from sediment traps (Pg. 2, lines 10 – 13) and have added additional mentions in the abstract.

Reviewer comment: Section 1.2: The authors should shortly mention the problem of collecting live pteropods ("swimmers") in sediment traps and how that could have

affected their work approach and results. (Alternatively it might be mentioned on P4 last paragraph).

Author response: Good point - we have added discussion of swimmers entering sediment trap samples in the introduction (Pg. 5, lines 1 – 4), and have mentioned how our results may have been impacted by swimmers in the discussion (Pg. 12, lines 24 – 30). Introduction: "A further complication of sediment trap data is that interpretation can be skewed by the presence of 'swimmers', i.e., specimens that were alive when they entered the trap (Harbison and Gilmer, 1986). This is a particular concern with pteropods as they sink to avoid predation (Harbison and Gilmer, 1986) and therefore may enter into the trap while still alive." Discussion: "These results could have been further complicated by the presence of swimmers, which would have entered the trap live and therefore would not have undergone any dissolution in the water column. If there was an increase in swimmers entering the traps at one time of year relative to another, it could be interpreted as less water column breakdown during these months. The most pristine shells in this study entered the trap in June and December, suggesting that there was not a seasonal pattern to swimmer frequency. We therefore assume that the number of swimmers entering the sediment trap is constant throughout the year and therefore does not affect the seasonal trends reported above."

Reviewer comment: Throughout the ms, they need to make clear whether they talk about live or dead organisms.

Author response: We agree that the distinction between processes that affect live and dead shells was not clear in the original manuscript. We have worked to clarify how different processes affect live, dead, and preserved shells: Abstract: Pg. 2, lines 10 – 13, Introduction: Pg. 4, line 33 – Pg. 5, line 11, Discussion: Pg. 12, lines 5 – 24.

Reviewer comment: P4L6: Lischka and Riebesell 2017 (Polar Biol, Volume 40) also studied metabolic response of pteropods (oxygen consumption).

Author response: Thank you for reminding us of this work - we have added the reference (Pg. 4, lines 11 – 12)

Reviewer comment: P4L24: through misses the "r"

Author response: corrected

Reviewer comment: P5L33: remove comma between body and whorl

Author response: In molluscs, the final whorl is also known as the body whorl. The sentence is grammatically correct as it was, "...the final, or body, whorl." (Pg. 6, line 17)

Reviewer comment: P6L10: Please detail at what temperature and for how long shells were dried.

Author response: We have added details about drying to the methods section (Pg. 6, lines 28 – 29): "Calcareous plankton were wet-picked, and left to dry in a 40°C oven for 24 hours, before being, dried, and stored for faunal analysis (E.Tappa pers.comm.)."

Reviewer comment: P10L31/32: Could any changes detected originate in the time prior collection in the trap during live in the water column?

Author response: This is an interesting question. We assume that there was no in-life dissolution as the water is permanently supersaturated with respect to aragonite, and because we don't see any evidence of patchy dissolution, such as is seen when pteropods undergo dissolution in-life (e.g. Peck et al., 2016, 2018). We have clarified this in the text (Pg. 12, lines, 7 – 11): "The water in the Cariaco Basin was supersaturated with respect to aragonite throughout the study. The thin aragonite shells of the pteropods are therefore chemically stable in the water column so it is unlikely that they underwent in-life dissolution. Furthermore, there is no evidence of patchy dissolution in pristine shells, or those which have undergone dissolution, such as has been observed in pteropod shells undergoing in-life dissolution in naturally undersaturated environments (Peck et al., 2016; 2018)."

Reviewer comment: P11L7/8: How can the authors know, pteropods were dead already? How likely is it that shell deterioration happened on the live organism? The assumption that any shell degradation took place only when organisms were dead already, is this simply based on the assumption that under aragonite supersaturated conditions no shell deterioration happened? If so, state clearly and support your view.

Author response: See response above

Reviewer comment: P11L20: : : : in the overall trend: : : (remove "is no")

Author response: done

Reviewer comment: Fig. 4: It would help the clarity of the figure if September, June, December (mentioned in the text) could be indicated on the x-axis.

Author response: Good point – see updated figure (now figure 5)

Reviewer comment: Fig. 8: Italics for Heliconoides inflatus

Author response: changed

We thank Reviewer #1 for their thorough and constructive review of our manuscript. Their comments have helped us to improve the scope and clarity of our work

[Figure]

**Fig. 1.** Figure 5: Shell condition over the study period

---

## Author Comment (AC2) · 22 Jan 2020

Rosie L. Oakes and Jocelyn A. Sessa

roakes@drexel.edu

Received and published: 22 January 2020

Reviewer comments have been copied below and author responses are listed underneath them. Page and line numbers refer to the tracked changes version of the manuscript.

Reviewer #2 - K. Kimoto

This manuscript describes the biometrics of pteropod shell and its degradation based on sediment trap samples in the Cariaco Basin. This kind of works of pteropod shells

using the sediment trap samples are insufficient, so it is very important to trace the biological responses to the ocean acidification and related ocean environmental changes. Especially the information of tropical species is less. In this sense, this work has the potential to become the base and develop criteria for this kind of study. Below I pointed out some concerning issues for this ms and make the comments

Reviewer comment: The relationship between shell length and whorls. According to photos in supplemental document, the aperture of some shells was damaged or showed bad preservation. The author inferred that shell diameter and whorl does not show clear relationship, but if a part of aperture had lost by dissolution and/or physical damage, its relationship between shell length and whorl might become uncertainty. If the plankton tow samples are available, the authors should use those plankton samples, not sediment trap ones. Or at all possible, the authors should use the only perfect shell in order to interpret length-whorl relationship.

Author response: Thank you for pointing out this potential source of bias. The fragility of H. inflatus shells make them particularly susceptible to breakage, often during collection (Pg. 7, lines 7 – 8). Shell diameter and shell whorl measurements were made on CT-scanned specimens but analyses were also run on a subset of 29 unbroken specimens. There was a weak but statistically significant relationship between shell diameter and the number of whorls. The FDR corrected p value was identical in both the whole and the subset dataset, although the R2 was higher in the subset dataset (whole dataset: R2 = 0.074, p Bon. = 0.415, p FDR = 0.057; subset dataset (Table S1): R2 = 0.101, p Bon. = 0.513, p FDR, 0.057). We have added this information into the methods (Pg. 7, lines 8 – 9), and results (Pg. 10, lines 27 – 30) sections of our manuscript. Since there was no difference between the two analyses, Figure 6 in the main manuscript contains the full dataset. The same figure with only the unbroken specimens is reproduced in the supplemental materials (Fig. S4).

Methods: "Because H. inflatus shells are fragile, they often break at the aperture during collection and processing. Although shell diameter and number of whorls were mea-
sured on all CT-scanned specimens, a subset of 29 shells with complete apertures was created for further analyses (Table S1)."

Results: "There was a weak, but statistically significant correlation between shell diameter and the number of whorls which remained when analyzing the subset of complete shells (Table S1) (whole dataset: R2 = 0.074, p Bon. = 0.415, p FDR = 0.057; subset dataset (Table S1, Fig. S4): R2 = 0.101, p Bon. = 0.513, p FDR, 0.057)."

Reviewer comment: Shell dissolution: How and when? The authors described that preservation states of shells in the sediment trap was not related to the duration time, so it might be negligible dissolution in the sediment trap collection cups. If this is correct, shell dissolution occurred at the water column, and it was associated with micro-zoo/bacterial activity which was decomposing organic tissues. My questions are that 1) in this case, does shell dissolution occurred at the inside, and outside of shell is sufficiently preserved? Can the authors show this evidence? Based on the photographs on the supplement material, surface texture of some shells looks like cloudy and lost their luster, indicating dissolution of outer shell. I am wondering that the decision of less dissolution in the sediment trap collection cups based on the result of relationship between residence time and LDX might be insufficient. In other words, I infer that shell dissolution occurs not only by microorganisms/bacterial activity but also post depositional oxidization in the sediment trap cups, as authors mentioned. I am understanding that this certification is very difficult, but the authors are using SEM, so please show some possibilities from the direct observations of materials.

Author response: We agree that this is an important question and decided to investigate a subset of seven shells, ranging from the best to the worst shell condition, under the scanning electron microscope (Page 8, section 2.4, lines 1 - 7). Our investigations revealed that the majority of the dissolution at LDX scores below 2.5 occurs on the outside of the shell. At LDX scores of 2.5 or higher, there is dissolution on both the inside and the outside of the shell. The external dissolution could be attributed to either dissolution associated with decaying organic matter in the water column, or alteration
associated with the preservative in the sediment trap cup. Internal dissolution is associated with the decaying organic body of the pteropod and/or alteration in the sediment trap cup. We have added discussions of these possible scenarios in the manuscript (Pg. 12, lines 18 - 19): "Scanning Electron Microscopy reveals that the majority of this dissolution occurred on the outside of the shells (Fig. 2, Fig. S2).", and (Pg. 13, lines 2 -3): "SEM images reveal that the internal shell walls were only impacted by dissolution at LDX values of 2.5 and higher (Fig. 2 d, h, l), indicating that the preservative did not cause dissolution." We have added a figure of a selection of the SEM images to the main manuscript (Fig. 2) and all the SEM images to the supplemental (Fig. S2).

Reviewer comment: Another possibility, is it available the comparison between organic carbon of the samples and LDX? Highly input of organic carbon flux induce carbonate dissolution at the inside of sediment trap cup.

Author response: Unfortunately, there are no measurements of the organic carbon content of the samples from the sediment trap cups. Particulate Organic Carbon was measured as part of the CARIACO time series hydrographic measurements; however, these values are only available for the first half of this study so we cannot make this comparison.

Reviewer comment: Relating to above, I understood this study is the first, and make the baseline of this kind of pteropod study, but it is bit unclear the main subject and purposes. If the authors interpretation is correct, does the pteropod shell of this species /or in this region not become an index of ocean acidification? I suppose that the authors want to make the criteria as OA index by using pteropod shell, but in this case, I think that shell preservation states indicate microorganisms activity in the pteropod shell.

Author response: Hopefully the restructuring of the abstract, and introduction have served to clarify the purpose of this study. From an ocean acidification perspective, we make the point that although many studies have focused solely on aragonite saturation, the availability of food and the collection and preservation methods used may also
affect shell condition (Pg. 14, section 4.3). We hope this research leads to a more holistic view of shell condition interpretations. We have rephrased the conclusion to clarify this point (Pg. 15, lines 27 - 32): "This demonstrates that in this aragonitesupersaturated setting, the availability of food has a greater control on shell formation than aragonite saturation. This pattern has been seen in other groups of molluscs, such as oysters and mussels and underlines the necessity of assessing pteropod shell parameters and dissolution in the context of multiple biotic and abiotic factors, not just aragonite-saturation. We hope that the baseline dataset of pteropod shell parameters presented in this study is the first of many focused regional studies around the world. These datasets will enable the quantification of the response of this sentinel group to ocean acidification."

Reviewer comment: I think it is very important finding that shell thickness does not have relationship with surrounding omega value (but still supersaturated). I strongly agree with the authors that they have resistance characteristics to small changes of saturation states and depends on available food to build their shells. However, the authors did not show what kind of food is important for their prey. If their main food is phytoplankton, please show their annual variations through a year instead of nutrient concentrations (Or is it possible to show the number of diatom bulbs in the sediment trap cups?) Because their food is particulate matters, not chemical component. It might be a good evidence to indicate their food availability.

Author response: Yes, good point. Although we don't have any diatom counts from these sediment trap cups, we have added references to two other studies conducted in the same basin which find that both organic carbon production (Thunell et al., 2000) and diatom populations (Romero et al., 2009) show strong increases at times of upwelling. Nutrient concentrations are good proxies for upwelling and therefore showing the nutrient changes through the year is a reasonable approximation for food availability (Introduction: Pg. 5, lines 21 - 24, Discussion: Pg. 14, lines 13 - 16).

Introduction: "Organic carbon fluxes in the basin vary in response to these hydro-

BGD
graphic changes, with one study reporting a tripling of primary productivity in response to upwelling (Thunell et al., 2000). Diatoms, a known food source for pteropods (Lalli and Gilmer, 1989), contribute to over 50% of this organic carbon flux, with their blooms coinciding with hydrographic and nutrient changes during times of upwelling (Romero et al., 2009)."

Discussion: "These upwelling-related nutrient changes in the Cariaco Basin have been shown to correspond with increases in organic carbon flux and diatom blooms (Thunell et al., 2000; Romero et al., 2009), indicating that pteropod food supply (Lalli and Gilmer, 1989) increases during upwelling conditions."

Pteropods eat diatoms, as well as dinoflagelletes and tintinnids. We have added a reference about this to the introduction (Lalli and Gilmer, 1989) (Pg. 4, lines 3 - 4): "Pteropods are also key components of the marine food web, feeding on phytoplankton and small zooplankton, such as diatoms, dinoflagellates, and tintinnids (Gilmer and Harbison, 1986, 1991; Lalli and Gilmer, 1989)".

Reviewer comment: The authors did not touch the phylogenetic variation of the species, but I am wondering the possibility of mixture of some lineages of this species. H. inflatus is certificated as a single-genetic species around the Cariaco basin? Or exists some cryptic species? If the author has this kind of information, please mention it for just confirmation. It is possible that the plasticity of shell (shell length, number of whorls) of this species that author mentioned is related with the phylogenetic variations.

Author response: There has not been any genetic work done on H. inflatus. Both Van der Spoel (1967) and Janssen (2004) noted that there was variability in the shape and location of the rib, however, there is no data to determine whether this in intra- or interspecific variability. We have added this information to the discussion (Pg. 13, line 34 – Pg. 14, line 4): "Both Van der Spoel (1967) and Janssen (2004) have described variability in the shape and position of the aperture tooth in H. inflatus, which could be attributed to intraspecific or interspecific variations. As there has not been any genetic
work conducted on H. inflatus from the Caribbean, we cannot be sure that the variability we see in shell shape cannot be attributed to two or more genetically-defined species."

Reviewer comment: Can the author interpret about morphological implication from the microXCT analysis Because it is very powerful tool and shows huge possibility for morphological information. If the authors want to indicate some suggestive issues, please make comment for following researchers and future study.

Author response: We agree that the CT data offer the opportunity to do some really interesting geometric morphometric work. We have added a section entitled "Further Work" (Pg. 15, lines 1 - 9) where we discuss the challenges of the field of gastropod geometric morphometrics, and discuss two recent studies. Our CT data will be available on publication and so can be used in future geometric morphometric studies.

We would like to thank Dr. Kimoto for his constructive review. It has helped us improve the completeness and clarity of this work.

**BGD**
**ope and SEM images s C8**

---

## Author Response (AR2)

Dear Dr. Gattuso,

Thank you for reviewing the changes we made to our manuscript entitled, "Determining how biotic and abiotic variables affect the shell condition and parameters of *Heliconoides inflatus* pteropods in the Cariaco Basin". We are delighted that the paper has been accepted for publication in Biogeosciences pending our response to the following comments:

**1) In response to my review the authors changed the title, however, they did not include mention that their study deals with sediment trap samples. I still think it is worth mentioning this in the title and would like them to include this.**

Based on previous comments from Reviewer #1, we changed the title of our manuscript from "Assessing annual variability in the shell thickness of the pteropod *Heliconoides inflatus* in the Cariaco Basin using micro-CT scanning",  to "Determining how biotic and abiotic variables affect the shell condition and parameters of *Heliconoides inflatus* pteropods in the Cariaco Basin". Because this revised title is longer than the original, we opted to not include "shell dissolution index, size and thickness as revealed from sediment trap samples" in the revised title as proposed by Reviewer #1 because we did not want to discourage a broad readership from this paper by making the title both too long and too specific. We feel the current title, "Determining how biotic and abiotic variables affect the shell condition and parameters of *Heliconoides inflatus* pteropods in the Cariaco Basin" accurately describes the work conducted in this paper and would like for this to remain the title of this paper. However, if, as the editor, you think that including 'from a sediment trap' in the title is of critical importance, we suggest, "Determining how biotic and abiotic variables affect the shell condition and parameters of *Heliconoides inflatus* pteropods **from a sediment trap** in the Cariaco Basin"

**2) In the abstract, L11+12, I think they could be a bit more precise here. The three stages where dissolution can happen are correct, but depending on the water mass, OA could also play a role during stages 2 and 3.**

We agree that OA could play a role during stage 2 and have changed point 2 in the abstract to read, "2) when organisms are dead in the water column associated with biotic decay of organic matter **and/ or abiotic dissolution associated with ocean acidification**". Because previous work by Oakes et al. (2019, Global Biogeochemical Cycles) found that the majority of pteropod dissolution after death was caused by the biotic decay of organic matter, we have also expanded the discussion of dissolution of carbonate in the water column to clarify these findings. Page 4 lines 30 – 34 now read: Organisms falling through the water column may decay en-route to the sediment trap, which can cause dissolution in calcareous organisms (Lohmann, 1995; Milliman et al., 1999). **In pteropod shells specifically, Oakes et al. (2019a) found the majority of post-mortem dissolution was associated with the biotic decay of organic material on the inside of the shell,** and therefore specimens from sediment traps do not perfectly capture in-life shell conditions.

We do not agree that OA could play a role in shell dissolution in stage 3. At that point, pteropod shells are in sealed sediment trap cups that were filled with preservative prior to sediment trap deployment. As the shells are not in contact with the seawater once they are in the sediment trap cup, but rather are within preservative fluid, the chemistry of the seawater could not affect their shell condition. We have added the word 'closed' in front of sediment trap cup to clarify that these samples are isolated from the seawater, "3) when organisms are in the **closed** sediment trap cup associated with the abiotic alteration by the preservation solution.".

We hope you find our responses acceptable. Please do not hesitate to get in touch if you have any further questions.

All the best,

Rosie Oakes

[revised manuscript text omitted]